# Identifying candidate items for a health-related quality of life measure in young children with respiratory illness: A scoping review of generic and disease-specific measures

**Michaile Gizelle Anthony**[1]*, **Graeme Hoddinott**[1,2], **Dzunisani Patience Baloyi**[1], **Anneke Catharina Hesseling**[1], **Marieke Margreet van der Zalm**[1]

**1** Desmond Tutu TB Centre, Department of Paediatrics and Child Health, Faculty of Medicine and Health Sciences, Stellenbosch University, South Africa, **2** School of Public Health, Faculty of Medicine and Health, The University of Sydney, Australia

* manthony@sun.ac.za

## Abstract

### Background

Health-related quality of life (HRQoL) in young children affected by respiratory illnesses remains understudied, particularly in low- and middle-income countries (LMICs), where the burden of these diseases is disproportionately high. Existing HRQoL measures, including both generic and respiratory disease-specific measures, have been reviewed to identify key components that can guide the development of a novel HRQoL tool for young children (0–5-years) with respiratory illnesses. The study aimed to identify candidate items from existing HRQoL measures to inform the development of a new HRQoL tool for young children (0–5 years-old) with respiratory illnesses in a LMIC setting.

### Methods

A scoping review was conducted using PubMed, EBSCOhost, and PsycArticles databases Keywords included variations of the following terms 'quality of life', 'health-related quality of life', 'wellbeing', 'questionnaire', 'instrument', 'measure', 'children', 'toddler', 'paediatric', 'child*', 'develop' or 'validation'. The search was limited to English-language articles published between January 2000 and November 2023. Deductive thematic analysis was used to organise the measures and synthesise cross-cutting components.

### Results

Out of 1823 articles, data were extracted from 72 articles reporting on 41 measures. Of these, 20 were generic, and 21 were specific to respiratory diseases. The

**Data availability statement:** All relevant data are within the paper and its Supporting Information files.

**Funding:** Marieke van der Zalm and Michaile Anthony are supported by a career development grant from the EDCTP2 programme supported by the European Union (TMA2019SFP-2836 TB lung-FACT2) and Marieke van der Zalm is also supported by the Fogarty International Center of the National Institutes of Health under Award Number K43TW011028. The content is solely the responsibility of the authors and does not necessarily represent the official views of the National Institutes of Health. Graeme Hoddinott's writing time was covered with financial assistance from the European Union (Grant no. DCI-PANAF/2020/420-028) through the African Research Initiative for Scientific Excellence (ARISE) pilot pro-gramme. ARISE is implemented by the African Academy of Sciences with support from the European Commission and the African Union Commission. The contents of this document are the sole responsibility of the author(s) and can under no circumstances be regarded as reflecting the position of the European Union, the African Academy of Sciences, and the African Union Commission.

**Competing interests:** The authors have declared that no competing interests exist.

measures' key dimensions included physical and emotional health, social support, and school functioning. However, few measures, targeted children 0–5-years, and none incorporated child-specific methods for assessing HRQoL. Existing tools varied widely in their domains and definitions, often lacking consistency and not adequately considering the developmental milestones. Furthermore, most tools were developed in high-income settings (HIC), with limited adaption to the socio-economic burden and disease burden contexts in LMICs.

## Conclusion

There is an urgent need for a comprehensive HRQoL measure tailored to young children with respiratory illnesses, particularly one designed for use in LMICs. Such a tool should address developmental milestones, cultural sensitivity, and the unique socio-economic challenges faced in these settings.

## Introduction

Lower respiratory tract infections (LRTIs) are the leading cause of paediatric hospi-talisations worldwide, with an estimated 11.9 million children <5-years hospitalised in 2019 [1–5]. Viral infections account for approximately 90% of all paediatric LRTIs [5], but bacterial infections, inflammatory conditions, and structural abnormalities also contribute to significant morbidity. The burden of these illnesses is particularly high in low- and middle-income countries (LMICs), where access to healthcare, early diagnosis, and appropriate management can be challenging [5].

Respiratory illnesses can have long-term impacts on young children, extending beyond the acute phase. Early-life LRTIs are associated with later conditions such as asthma and potentially chronic obstructive pulmonary disorder (COPD). Children who experience recurrent or severe respiratory illnesses may face ongoing respira-tory symptoms, developmental delays, and psychosocial challenges, affecting their overall well-being [6–8]. Existing data on the impact of respiratory illnesses and per-ceived HRQoL are particularly scarce in LMICs, where the burden of these diseases is highest [9–11]. In this study, respiratory illness refers to any condition affecting the airways, lungs, or related structures, causing symptoms like coughing, wheezing, shortness of breath, or respiratory distress. This includes infections, inflammatory conditions, and structural abnormalities.

Despite the profound health and social implications of respiratory illnesses, cur-rent tools do not capture their impact on young children's health-related quality of life (HRQoL) [9–11]. HRQoL is a multidimensional concept that encompasses physical health, emotional well-being, and social interactions, all of which are critical for early childhood development [12].

There is a notable gap in HRQoL measurement for young children with respira-tory illnesses, particularly in LMICs, where the burden is highest [9,10]. While some disease-specific HRQoL measures exist—such as those for asthma and cystic fibrosis [13]. HRQoL measures provide valuable tools for clinicians to evaluate

treatments, tailor interventions, and improve care. They also inform health policy and resource allocation [12,14]. These tools can be generic, applicable across populations, or disease-specific, focusing on particular symptoms or conditions [15,16]. For children with respiratory illnesses, challenges such as decreased energy, sleep disturbances, and social stigma are often reported [11]. Existing data on the impact of respiratory illnesses and perceived HRQoL are particularly scarce in LMICs, where the burden of these diseases is highest, yet no existing HRQoL measure comprehensively addresses these issues for this age group [9,11].

Although disease-specific HRQoL measures exist, such as CFQ-R for cystic fibrosis, PedsQL Asthma, TACQoL Asthma etc. [17–20], they are often overly specific and not suitable for general use across respiratory illnesses [10]. Moreover, these tools have limitations when applied to high-burden, resource-constrained LMICs, where considerations of developmental milestones, cultural sensitivity, and socio-economic factors are critical [9,10].

This review provides an updated overview of HRQoL measures, with a particular emphasis on tools relevant to young children in LMICs. It builds on earlier reviews such as those by Quittner et al., (2008) and Kwon et al., 2023, by identifying critical gaps in HRQoL measurement for specific respiratory illnesses, including pulmonary TB (PTB), which is prevalent in LMICs. These critical gaps include the focus on young children (0–5-years-old), identifying gaps in existing HRQoL measures, addressing the LMIC as well as highlighting the need for a comprehensive approach to the development of HRQoL measure for young children 0–5-years-old in LMICs.

This study aimed to identify candidate items from existing HRQoL measures to inform the development of a new HRQoL tool for young children (0–5 years-old) with respiratory illnesses in a LMIC setting, addressing a significant gap in the current literature.

## Methods

A scoping review was conducted, including both published peer-reviewed and grey literature. The review followed the five stages outlined in Arksey and O'Malley's framework, supplemented by the Joanna Briggs Institute 2015 recommendations [21,22]. Reporting adhered to the preferred reporting items for systematic reviews and meta-analyses extension for scoping reviews (PRISMA-ScR) [23].

### Stage 1: Identifying the research question and defining the study scope

The primary aim was to identify all HRQoL measures, including both generic and disease-specific tools, developed for use in children, with a specific focus on measures designed for respiratory illnesses. The review included literature published between January 2000 and November 2023, restricted to English-language manuscripts. The most recent search conducted was on the 30th of November, 2023.

The inclusion criteria encompassed studies reporting on the development, validation, or adaption of HRQoL measures for children. Excluded were conceptual research, studies that applied HRQoL measures without reporting on their development or properties, review articles, and articles focused exclusively on adolescents ≥18-years or adults. The full inclusion and exclusion criteria are outlined in Table 1.

### Stage 2: Identify relevant studies

An initial search of PubMed was conducted to identify relevant articles. This involved analysing the title and abstract text as well as the MeSH terms used to index the articles. Used to describe the articles. Based on these findings, the search terms were refined to focus on quality of life (QoL) and children. The inclusion and exclusion criteria was refined to focus exclusively on respiratory illnesses, excluding research on other diseases and review articles. This targeted approach was intended to provide a cohesive understanding of HRQoL specifically within the context of respiratory conditions, enabling us to underscore the unique challenges and unmet needs in this population. A revised search was undertaken across

**Table 1. Inclusion and exclusion criteria.**

| Included |
| --- |
| • Research articles that described, developed, and/or evaluated HRQoL measurements for children aged 0–18-years-old. |
| • Research articles that described generic, respiratory-disease-specific measures, self-reports, and parent proxy measures. |
| • Research articles that highlighted the domains and items of the HRQoL measure. |
| • Research articles specifically focused on the development and/or validation of respiratory-disease-specific measures. |

| Excluded |
| --- |
| • Research articles that were written in a language other than English. |
| • Research articles published before 2000. |
| • Research articles that reported HRQoL using a tool but did not report any psychometric or tool development data on the tool itself were excluded. |
| • Disease-specific tools unrelated to respiratory illnesses. |
| • Review articles were excluded. |
| • Research articles focusing on cases requiring mechanical intervention or presenting with severe respiratory compromise, which includes children with tracheostomies and those using non-invasive ventilation due to muscle weakness from neuromuscular diseases were excluded. |

PubMed, EBSCOhost and PsycArticle. Keywords included a variation of the following terms 'quality of life', 'health-related quality of life', 'wellbeing', 'questionnaire', 'instrument', 'measure', 'children', 'toddler', 'paediatric', 'child*', 'develop' or 'validation'. The full search strategy is provided in (S1 Appendix).

A faculty librarian and senior authors (GH & MVDZ) were consulted and assisted in refining the search terms. The iterative search strategy included a broad range of terms such as "health-related quality of life," "HRQOL," and "functional status" to ensure comprehensive coverage of relevant dimensions of QoL assessments.

### Stage 3: Study selection: the process of identifying and screening records

The results, including titles, and abstracts, were exported and duplicates were removed. The initial screening was performed by the first reviewer (MA), who excluded records unrelated to the development, evaluation, or validation of QoL measures. Two reviewers MA & GH independently screened the remaining results by title. Discrepancies were resolved by a 3rd reviewer MVDZ.

Subsequent abstract screening was conducted by MA & MVDZ, with conflicts resolved by GH. Based on the initial review, the inclusion and exclusions criteria were refined in consultation with senior authors (GH & MVDZ) to focus exclusively on respiratory conditions. Articles unrelated to respiratory conditions, including reviews and studies on other diseases, were excluded to maintain coherence and relevance.

Two reviewers (GH & MA) independently reviewed the full text of eligible articles. Discrepancies were resolved in consultation with MVDZ. This rigorous screening process ensured that only full-text, peer-reviewed articles were included in the scoping review. Additionally, grey literature was limited to full-text, peer-reviewed articles that were not identified through database searches, ensuring a comprehensive synthesis of the existing evidence on QoL in respiratory illnesses among children.

### Google search engine and reference list searches

A grey literature search was conducted using the Google web browser. Keywords and inclusion criteria were applied, and results were limited to the first ten pages of search results. A screening process similar to that of the database searches was followed. Reference lists of systematic reviews and included full-text articles were manually screened for additional studies.

Reference lists of systematic reviews and articles that were identified through the database search and Google search were manually screened. Reference lists of the full-text research articles included in our analysis were also screened for additional articles to include in the review. A comprehensive review was conducted by systematically searching

scientific literature, supplemented with grey literature sources to capture a broader and more inclusive range of relevant information.

### Stage 4: Charting the data

Data extraction was conducted manually using a pre-piloted charting table in MS Excel.

The charting table included data on the study authors and year of publication, generic or disease-specific measures, domains measured, target population (including age), self-report or parent proxy report, response options, and language.

The researchers intentionally did not start with a predefined list of domains, as they wanted to be as comprehensive as possible and allow for the natural emergence of relevant domains during extraction. After gathering the data, we grouped the individual domains based on related attributes—for example, all aspects related to mobility, such as physical functioning, running, and walking, were categorised under the broader physical domain. This approach allowed us to maintain rigour and ensure that our findings captured the full breadth of relevant measures, with the senior authors verifying our groupings to uphold the study's integrity.

### Stage 5: Collating, summarising, and reporting the results

The articles were thematically grouped into generic and disease-specific measures. Within disease-specific measures, articles were further categorised by disease areas including bronchopulmonary dysplasia, cystic fibrosis, asthma, primary ciliary dyskinesia, chronic cough, sinus, upper respiratory tract infection, and rhinitis.

Measures were also grouped based on the applicable age range with tools for children 0–5-years priortised in the presentation. Subsequently, the broad age ranges were highlighted. Articles were summarised based on HRQoL domains reflected, respondent type (e.g., parent proxy, child self-report measures), and country of origin and/or translation. This iterative process was conducted in discussion with senior authors GH and MVDZ, ensured a comprehensive and structured analysis of the data.

### Ethics considerations

The study were approved by the Health Research Ethics Committee at Stellenbosch University,

Tygerberg, South Africa (S21/05/084). Furthermore, the study was conducted in accordance with the Declaration of Helsinki.

## Findings

### Overview of the identified measures of HRQoL

The initial search yielded, 1823 peer-reviewed research articles, of which 495 were duplicates and subsequently removed (See Fig 1). The remaining 1340 unique articles were screened by title, leading to the exclusion of 877 articles. Of the 463 articles screened by abstract, 322 were excluded as they included disease-specific tools unrelated to respiratory illnesses. A total of 141 full-text articles were assessed for eligibility, with 92 excluded for the following reasons a) not being about the development and evaluation of an HRQoL measurement (n=45), b) review articles (n=44), and c) duplicates (n=3).

In total, 72 research articles met the inclusion criteria and were included in the full-text review. These studies described 41 HRQoL measures, including 20 generic and 21 disease-specific tools. Among these, five measures (four generic—IQI, TAPQOL, EQ-TIPS and ITQOL-SF-47 and one disease-specific measure—the Cystic Fibrosis Questionnaire-Revised) were developed specifically for children aged 0–5 years (Tables 2 and 3).

**Age range.** Of the 20 generic HRQoL measures, four were applied to children aged 2–18-years, five were designed for children aged 8–18-years, Seven were applicable to children 3–16-year, four were specifically developed for children aged 0–5-year (IQI, TAPQOL, EQ-TIPS and ITQOL-SF-47).

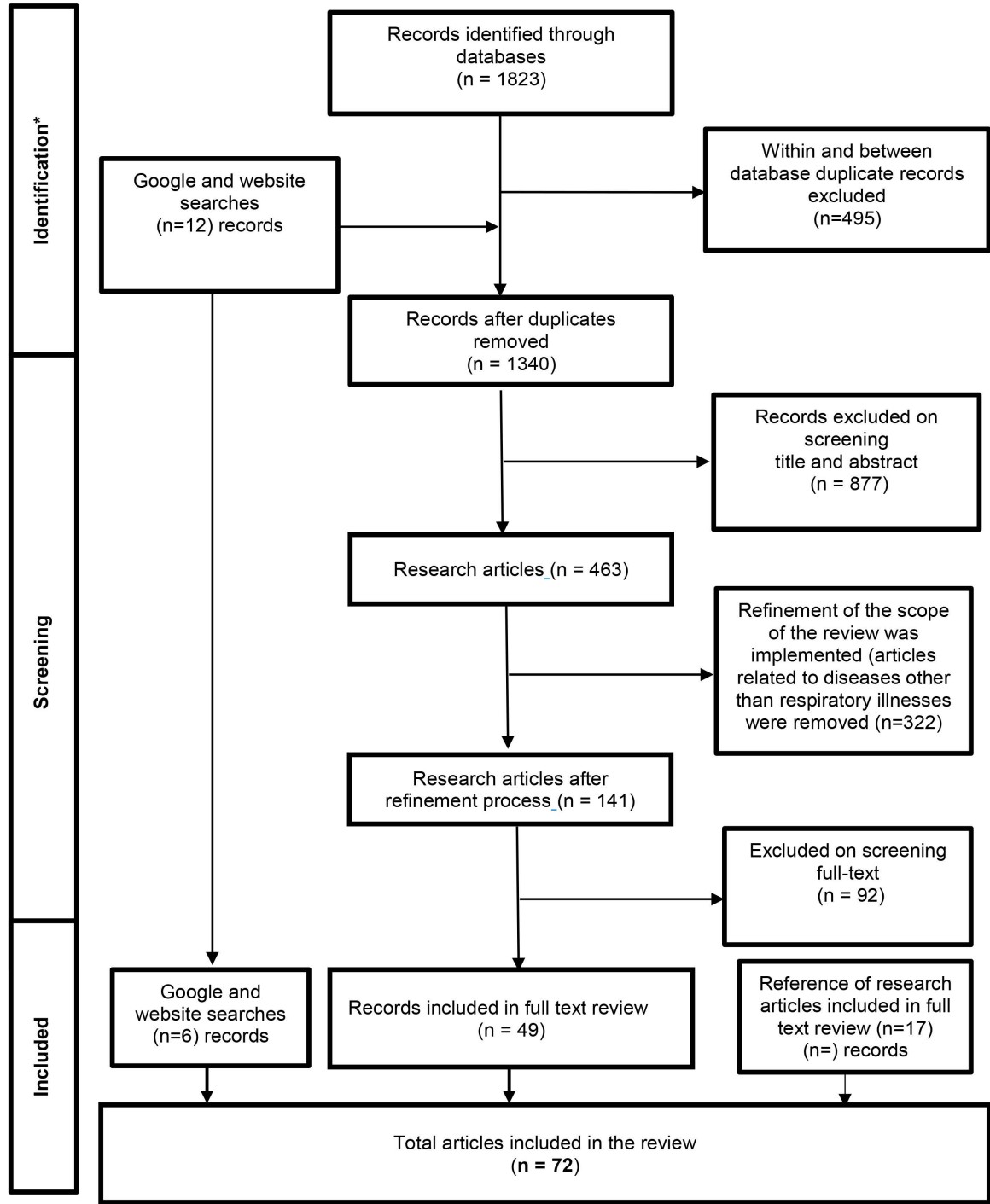

**Fig 1. Process of identification and screening of literature.**

**Table 2. Characteristics of the included generic measures of HRQoL.**

| # | Name | Age range | Respondent type | Nr domains | Nr of items | Dimensions/domains | Recall period | Measurement scale | Country developed Or validated | LMIC (yes/no) | Reference |
|---|------|-----------|-----------------|------------|-------------|--------------------|---------------|-------------------|-------------------------------|----------------|-----------|
| 1 | Infant health-related quality of life instrument (IQI) | 0-3 | Parent proxy | 7 | 7 | Sleeping, feeding, breathing, stooling/poo, mood, skin and interaction | Today | 4 point scale | Colombia, Hong Kong, United States of America, United Kingdom, China, New Zealand | No | [24,25] |
| 2 | TNO-AZL PRESCHOOL CHILDREN QUALITY OF LIFE (TAPQOL) | 0-5 | Parent proxy | 4 | 43 | Physical functioning, Social functioning, Cognitive functioning, Emotional functioning | Past 3 months | 3 point scale | Netherlands | No | [26] |
| 3 | DISABKIDS Smiley Questionnaire: The TAKE 6 Assisted Health-Related Quality of Life Measure | 4-7 | Assisted self-report & Parent proxy | 1 | 6 | Not reported | Not reported | 5 point scale | Austria, France, Germany, Greece, Netherlands, Scotland & Sweden | No | [27] |
| 4 | DISABKIDS-10 Index | 8-18 | Self-report and Parent proxy | 3 | 10 | Mental, Social and Physical impact of the heath condition | Past 4 weeks | 5 point scale | Portugal, | No | [28] |
| 5 | The DISABKIDS generic quality of life instrument (DCGM-37) | 4-16 | Self-report and Parent proxy | 6 | 37 | Independence, Physical limitation, Social inclusion, Social exclusion, Emotion, and Medication. | Not reported | 5 point scale | Austria, France, Germany, Greece, Netherlands, Scotland & Sweden, Japan | No | [29,30] |
| 6 | EQ-TIPS | 1-36 Months | Parent proxy | 6 | 6 | Movement, Play, Pain, Relationships, Communication, Eating | Today | 3 point scale | South Africa | Yes | [16,31–33] |
| 7 | The Quality of Life Scale for Children (QoL-C) | 4-9 | Self-report and Parent proxy | 5 | 5 | Moving (e.g., walking around), Looking after myself (washing or dressing myself), Doing usual activities (e.g., going to school, playing, hobbies, doing things with family/friends), Having pain (being sore), feeling worried, sad or unhappy. | Today | 3 point scale | England | No | [34] |
| 8 | EQ-5D-Y | 8-14 | Self-report and Parent proxy | 5 | 5 | Mobility ('walking about'), self-care ('looking after myself'), usual activities ('doing usual activities'), pain and discomfort ('having pain or discomfort'), and anxiety and depression ('feeling worried, sad or unhappy'). | Today | 3 point scale | Germany, Italy, SA, Spain, Sweden, Netherlands, United Kingdom | Yes | [35–37] |

*(Continued)*

| # | Name | Age range | Respon-dent type | Nr domains | Nr of items | Dimensions/domains | Recall period | Measure-ment scale | Country developed Or validated | LMIC (yes/no) | Reference |
|---|------|-----------|------------------|-----------|------------|--------------------|---------------|--------------------|-----------------------------------|---------------|-----------|
| 9 | Pediatric Quality of Life Inventory 4.0 Generic Core Scales | 2-18 | Self-report and Par-ent proxy | 4 | 23 | Physical, Emotional, Social, School | Past one month | 5 point scale | United States of America, Greece, Argen-tina, Iran, United Kingdom, Spain, Palestine, Fin-land, Malaysia) | No | [38–45] |
| 10 | PedsQL 4.0 SF15 | 2-18 | Self-report and Par-ent proxy | 5 | 15 | Physical functioning, Psychoso-cial health, Emotional function-ing, Social functioning, School functioning | Past one month | 5 point scale | United States of America | No | [46,47] |
| 11 | The Very Short Well-being Questionnaire for Children (VSWQ-C) | 6 and older | Self-report | 4 | 4 | Home life, School life, Friends, and Health | Past week | 5 point scale | England | No | |
| 12 | KINDL-R | 3-18- | Self-report and Par-ent proxy | 6 | 24 | Physical well- being, Emotional well-being, Self-esteem, Family, Friends, and Everyday function-ing (school or nursery school/kindergarten). | Past week | 5 point scale | Germany & Serbia | No | [48,49] |
| 13 | Kiddy-KINDL | 3-6 | Self-report and Par-ent proxy | 6 | 12 | Physical well-being, Psycho-logical well-being, Self-esteem, Family, Friends and Everyday Functioning. | Past week | 3 point scale | Germany | No | [50] |
| 14 | ITQOL-SF47 | 2 months-5-years-old | Parent proxy | 6 | 47 | Physical function, Growth and development, Bodily pain, Tem-perament and moods, Behavior, and General health perceptions. | Today | 4 point scale | Netherlands | | [51,52] |
| 15 | KIDSCREEN-10 | 8-18 | Self-report and Par-ent proxy | unidimen-sional global HRQoL index | 10 | Health | Last week | 5 point scale | Austria, Czech Republic, France, Ger-many, Greece, Hungary, Ireland, Poland, Spain, Sweden, Switzerland, the Netherlands, and the United Kingdom, Japan | No | [53,54] |
| 16 | KIDSCREEN-27 | 8-18 | Self-report and Par-ent proxy | 5 | 27 | Physical well-being, psycho-logical well-being, autonomy & parents, Peers & social support and School environment | Last week | 5 point scale | Japan | No | [54] |

*(Continued)*

**Table 2.** (Continued)

| # | Name | Age range | Respon-dent type | Nr domains | Nr of items | Dimensions/domains | Recall period | Measure-ment scale | Country developed Or validated | LMIC (yes/no) | Reference |
|---|------|-----------|------------------|------------|-------------|--------------------|--------------|--------------------|-------------------------------|---------------|-----------|
| 17 | KIDSCREEN – 52 | 8-18 | Self-report and Parent proxy | 10 | 52 | Physical well-being, Psychological well- being: life satisfaction and positive emotions, Moods and emotions, Social support and peers relation, Parents relation and home life, Self-perception: body image and self-esteem, Autonomy, Cognitive and school functioning, Bullying and social rejection, and Perceived financial opportunities. | Last week | 5 point scale | Austria, France, Germany, Spain, Switzerland, Netherlands, United Kingdom, Norway, Greece | No | [55–57] |
| 18 | FirstNations-CQOL | 0-12 | Parent-proxy | 3 | 21 | Patient experiences, Quality of life and Patient support | Not reported | Not reported | Australia | No | [58] |
| 19 | Multidimensional child well-being scale (MCWBS) | 9-15 | Self-report | 4 | 30 | Physical well-being, psychological well-being, social well-being and educational well-being | Not reported | 5 point scale | China | No | [59] |
| 20 | Kids-CAT | 7-17 | Self-report | 5 | 377 | Physical well-being, psychological well-being, family well-being, social well-being and school well-being | >4 weeks | Not reported | Germany | No | [60] |

Of the 21 disease-specific HRQoL measures, five measures were applicable to a broad age range of children aged 0–18-years. Fifteen were designed for children ≥4-years (see Table 3). Only one measure, the Cystic Fibrosis (CFQ Child version), was specifically tailored for children 0–5-years.

**HRQoL domains and items.** A total of 101 domains were identified across the 20 generic HRQoL measures, with the number of domains per measure from 1 to 10 (See Fig 2). The most frequently assessed domains across all ages included: 1) physical well-being (n = 14), 2) emotional well-being (n = 10), 3) psychological well-being (n = 8), and 4) social support/functioning (n = 13).

The number of items per measure ranged from 4 to 377, with considerable variability across domains. For children aged 0–5-years, commonly assessed domains included physical, emotional, cognitive, and social functioning. Less commonly measured domains in this age group included communication and sleep.

A total of 79 domains were identified across the 21 disease-specific measures, with the number of domains per measure ranging from four to nine. One measure, namely the Exeter health-related quality of life measure (EXQOL) did not report on specific domains. Commonly assessed domains across all ages included:1) symptoms (n = 12), 2) physical domain (n = 10), 3) emotional domain (n = 11), and 4) social domain (n = 8). The number of items per measure ranged from 5 to 68. For children aged 0–5 years, key domains included respiratory symptoms, treatment burden, vitality, health perceptions, and physical functioning (See Fig 3).

**Respondents to the HRQoL measures.** Among the 41HRQoL measures identified: 11 (5 generic; 6 disease-specific) were designed exclusively for parent proxy respondents (See Table 2 and Table 3). Nine (2 generic; 7 disease-specific) were exclusively for the use of child self-report HRQoL. The remaining 24 (12 generic; 12 disease-specific) were designed for both child self-report as well as parent proxy respondents. All five measures applicable to 0–5-years were designed for parent proxy respondents.

**Table 3. Characteristics of included disease-specific HRQoL measures.**

| # | Measure name | Disease Type | Age range | Respondent type | Nr domains | Nr of items | Dimensions/ Domains | Recall period | Measurement scale | Country developed/ validated | LMIC (Yes/no) | Reference |
|---|---|---|---|---|---|---|---|---|---|---|---|---|
| 1 | Child Chronic Cough quality of life measure (CC-QOL) | Cough | 7-17 | Self-report | 3 | 16 | 1) Physical domains, 2) Social domain & 3) Psychological domain | The past week | 7 point Likert scale | Brisbane | No | [61] |
| 2 | Parent-proxy Children's Acute Cough-specific QoL Questionnaire (PAC-QoL) | Chronic cough | 0-18 | Parent proxy | 3 | 16 | Physical, Social, and Emotional domains | 24 Hours | 7 point Likert scale | Brisbane, Melbourne | No | [62] |
| 3 | Paediatric chronic cough (PC-QOL) | Chronic cough | 0-14 | Parent proxy | 3 | 27 | Psychological, Physical and Social | Not reported | 7 point Likert scale | Brisbane | No | [63] |
| 4 | The Children's Health Survey for Asthma (CHSA-C) | Asthma | 7-16 | Self-report and Parent Proxy | 3 | 25 | Physical Health, Child Activities, and Emotional Health | 2 weeks | 5 point scale | United States of America | No | [64] |
| 5 | The Integrated Therapeutics Group Pediatric Asthma Short Form | Asthma | 5-12 | Parent proxy | 3 | 17 | Daytime symptoms, Night-time symptoms & Functional limitations | Past 4 weeks | 5 point scale | Not reported | Unknown | [65] |
| 6 | The Childhood Asthma Questionnaires (CAQ-B) | Asthma | 7-11 | Self-report and Parent Proxy | 4 | 23 | Not reported | Not reported | Not reported | Not reported | Unknown | [66] |
| 7 | "A pictorial version of the Pediatric Asthma Quality of Life Questionnaire (PAQLQ)" | Asthma | 5-7 | Parent proxy | 2 | 13 | Caregivers daily activities, Fear and worry | Past week | 7 point Likert scale | United States of America | No | [67] |
| 8 | PedsQL 3.0 Asthma Module | Asthma | 2-16 | Self-report and Parent Proxy | 4 | 28 | Asthma Symptoms, Treatment Problems, Worry, and Communication | past 1 month | 5 point scale | United States of America | No | [19,68] |
| 9 | The Exeter health related quality of life measure (EXQOL) | Asthma | 6-12 | Computer based Self report and parent proxy | Not reported | 12 | not reported | Not reported | 2 point scale | Not reported | Unknown | [69] |

*(Continued)*

**Table 3.** (Continued)

| # | Measure name | Disease Type | Age range | Respondent type | Nr domains | Nr of items | Dimensions/Domains | Recall period | Measurement scale | Country developed/validated | LMIC (Yes/no) | Reference |
|---|---|---|---|---|---|---|---|---|---|---|---|---|
| 10 | DISABKIDS Asthma Module (DISABKIDS-AsM) | Asthma | 8-18 | Self-report and parent proxy | 2 d | 11 | Impact and Worry | Not reported | 5 point scale | São Paulo | Yes | [70] |
| 11 | Asthma-related quality of life (ARQOL) | Asthma | 6-13 | Self-report | 5 | 35 | Restriction of social life, Physical disturbances, Limitation in physical activity, Daily inconveniences in managing the disease, and Emotional distress, | Not reported | Not reported | Taiwan | No | [71] |
| 12 | How are you? (HAY) | Asthma | 8-12 | Self-report and Parent Proxy | 4 Generic & 4 Asthma specific | 32 | **Generic domains:** Physical activities, Cognitive activities, Social activities, and Physical complaints **Asthma specific domains**: Asthma symptoms, Emotions related to asthma, Self-concept, and Self-management | previous week | 4 point scale | Holland | No | [72] |
| 13 | TACQOL-Asthma | Asthma | 8-16 | Self-report and Parent Proxy | 5 | 68 | Complaints (spontaneous asthma symptoms), Situations (that provoke symptoms), Treatment (visits to the doctors), Medication (use of) and Emotions (negative emotions). | Last month | 3 point scale (Child) 5 point scale (caregiver) | Netherlands | No | [20] |
| 14 | Cystic Fibrosis Questionnaire-Revised (CFQ-R) | Cystic Fibrosis | 4-60 Months | Self-report and parent proxy | 5 | 26 | Respiratory Symptoms, Treatment Burden, Vitality,. Health Perceptions and Physical Functioning were administered to parents of children ≥36 months due to limited relevance for younger children (e.g., "difficulty climbing stairs"). | Past week | 4 point scale | United States of America, Canada, Germany | No | [17,18] |
| 15 | Cystic Fibrosis Questionnaire (CFQ)–Child version | Cystic Fibrosis | 6-13 | self and parent proxy | 3 | 33 | Physical Symptoms, Emotional Functioning, and Social Functioning | Not reported | 4 point scale | United States of America and Puerto Rico | No | [73] |
| 16 | The Paediatric Allergic Rhinitis Quality of Life Questionnaire (Ped-AR-QoL) | Rhinitis | 6-14 | Self-report and Parent Proxy | 5 | 20 | Symptoms, Symptom duration, Emotion, Activities and Sleep | Last 2 weeks | 4 point scale | Greece | No | [74] |

*(Continued)*

**Table 3.** (Continued)

| # | Measure name | Dis-ease Type | Age range | Respon-dent type | Nr domains | Nr of items | Dimensions/ Domains | Recall period | Mea-sure-ment scale | Country devel-oped/ validated | LMIC (Yes/no) | Refer-ence |
|---|---|---|---|---|---|---|---|---|---|---|---|---|
| 17 | Health related quality of life instrument for Primary Ciliary Dyskinesia (QOL-PCD) | Primary ciliary dyski-nesia | 6-17 | Self-report and Parent Proxy | 5 | Child self-: 37 and Parent proxy ver-sion:41 | **Child self-report:** Phys-ical Functioning, Emo-tional Functioning, Social Functioning, Treatment Burden, Upper Respi-ratory Symptoms, Lower Respiratory Symptoms and Ears & Hearing Symptoms. **Parent proxy:** Physical Functioning, Emotional Functioning, Social Functioning, Treatment Burden, Upper Respira-tory Symptoms, Lower Respiratory Symptoms and Ears & Hearing Symptoms. Health Perception and Eating & Weight | Past week | 4 point scale | United States of America, United Kingdom, Ireland & Canada | No | [75] |
| 18 | "Wiscon-sin Upper Respiratory Symptom Survey for Kids (WURSS-K)" | Upper respi-ratory Illness | 10 and older | Self-report | 3 | 15 | Global illness severity, Severity-symptom and Impact | Today | 5 point Likert scale | Polish | No | [76] |
| 19 | Broncho-pulmonary dysplasia quality of life measure (BPD-QoL) | Bron-chopul-monary dyspla-sia | 4-8 | Parent proxy | 5 | 33 | 1) Pulmonary com-plaints, 2) school functioning, 3) Growth and nutrition 4) Exer-cise and locomotion & 5) Emotional func-tioning and health care concerns. | Not reported | 7 point Likert scale | Nether-lands | No | [77] |
| 20 | Sinus and Nasal Quality of Life Survey (SN-5) | Sinus | 2-12 | Parent proxy | 5 | 5 | Symptoms of infection, Nasal obstruction, Allergy, Emotional distress and Activity limitation | 4 weeks | 7 point Likert scale | United States of America | | [78] |
| 21 | Child acute cough-specific measures (PAC-QOL$_6$) | Cough | 0-18 | Self-report and Parent Proxy | 3 | 6 | Physical, social and emotional | Not reported | Not reported | Not reported | | [79] |

**Country of origin.** Of the 41 HRQoL measures: 24 (12 generic; 12 disease-specific) were developed in HICs (see Table 2 and Table 3). Two (1 generic; 1 disease-specific) measures were developed in LMICs including (South Africa and Sao Paulo). The EQ-TIPS (generic) was developed in South Africa. The DISABKIDS-AsM (disease-specific) was developed in Sao Paulo for children with asthma aged 8–18-years. Ten (7 generic; 3 disease-specific) measures were developed and validated in multiple country. Four disease-specific measures did not report the country of origin. The

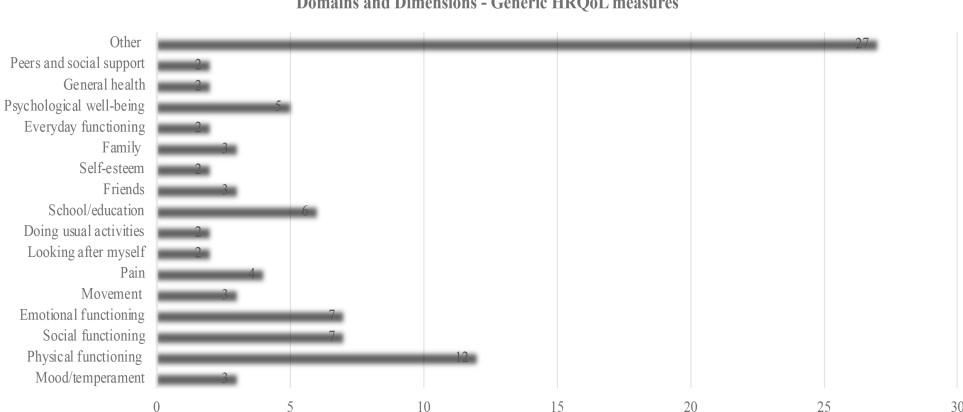

**Fig 2. Domains and Dimensions: Generic Measures.**

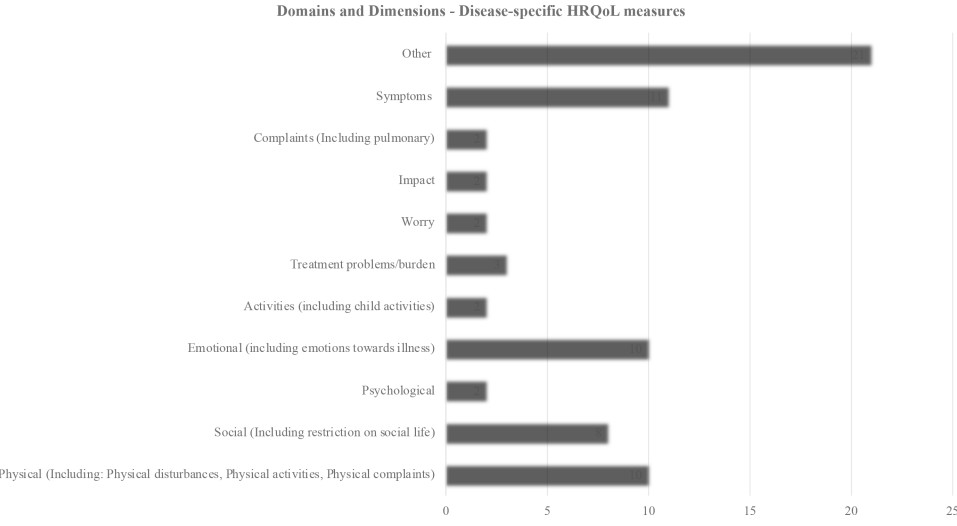

**Fig 3. Domains and Dimensions: Disease-specific Measures.**

development of the EQ-5D-Y took place in multiple countries, including SA, however, was not validated in a LMIC. Notably, none of the disease-specific measures for children aged 0–5 years were developed in LMICs.

## Discussion

This review screened 1,823 peer-reviewed articles, ultimately including 72 studies reporting on 41 distinct HRQoL measures, of which 20 were generic and 21 were disease-specific. Notably, only five measures—four generic and one disease-specific (for cystic fibrosis)—were designed for children aged 0–5-years. The identified HRQoL measures displayed substantial variability, encompassing 101 domains across generic measures and 79 across disease-specific measures. While the most commonly assessed domains included physical and emotional well-being, the majority of measures were developed in HICs, with only two originating from LMICs. This underscores a significant gap in contextually relevant HRQoL tools for children with respiratory illnesses, including PTB, in LMIC settings.

A key finding of this review is the limited availability of both generic and respiratory disease-specific HRQoL measures for young children aged 0–5-years. Furthermore, the development and validation of existing measures in LMICs remain scarce, despite the different socioeconomic contexts and high burden of respiratory diseases in these regions. While measures varied in their included domains and items, physical, emotional, cognitive, and social functioning were the most relevant domains for young children. However, many measures lacked developmental adaptations or clear guidance for assessing these domains in this age group.

Although valuable insights can be drawn from the available measures, several limitations must be addressed: 1) Existing tools exhibit inconsistent definitions, criteria, and inclusion of domains and items, limiting comparability across studies and 2;) Most HRQoL measures were developed in HICs and are not readily adaptable to LMICs without accounting for distinct cultural, linguistic, and socioeconomic factors. While HRQoL measures developed in HICs may not fully account for the unique challenges faced in LMICs, further research is necessary to assess the influence of cultural, linguistic, and socioeconomic factors on their applicability.

Research has demonstrated that HRQoL tools like the PedsQL or EQ-5D-Y can be successfully adapted for LMICs through rigorous cultural validation and iterative testing, such as cognitive interviews and pilot studies. These adaptations underscore the importance of contextualised approaches in HRQoL measurement. However, the lack of a disease-specific HRQoL measure tailored to young children with respiratory illnesses in LMICs—especially TB—remains a critical gap [10,80,81].

While previous reviews have focused on HRQoL tools for conditions like asthma, cystic fibrosis, vocal cord dysfunction, and sleep-related disorders [82]. This review serves as an update to this previous work, as it includes additional HRQoL measures related to respiratory conditions. There are now several well-documented and validated generic HRQoL measures for use in children and proxies, such as EQ-5D-Y [35–37], EQ-TIPS [16,32,33], and PedsQL [38,83]. Initially, only a few disease-specific measures developed, but these have now become more common and are available in several paediatric conditions such as asthma [19,20,64–71] and cystic fibrosis [17,18,73]. There has been growing interest in the assessment of HRQoL among children affected by respiratory illnesses including TB, in LMIC due to the potential long-term impact of disease on children's lives. However, there remains a lack of a disease-specific HRQoL measure for the youngest children with respiratory illnesses in an LMIC.

This review highlights substantial inconsistencies in the domains and items included in HRQoL measures, particularly those for young children [12,15]. The developmental stages of early childhood necessitate tools that not only assess functioning but also consider developmental milestones. Proxy reporting, commonly used for young children, poses challenges due to caregivers' limited ability to interpret non-observable aspects of HRQoL and the potential influence of their emotional proximity to the child [10]. Alternative methods, such as creative play, body mapping, and role-playing, may provide more nuanced insights into HRQoL for children aged 3–5-years.

This review has several strengths and limitations. A comprehensive search strategy captured all available data on HRQoL measures for children, providing a valuable resource for researchers. However, restricting the search to English-language articles published from 2000 onward may have excluded foundational studies or non-English research, particularly from LMICs. Additionally, limiting the review to English-language articles may have resulted in the omission of relevant research conducted in other languages, particularly from LMICs where diverse cultural and contextual factors significantly impact children's HRQoL. As a result, this may affect the comprehensiveness of our findings and limit the generalisability of the conclusions drawn from the available literature. Additionally, the exclusion of HRQoL websites and conference proceedings may have omitted emerging tools or insights not yet published in peer-reviewed literature. Consequently, some recent developments in HRQoL measures for paediatric respiratory conditions may not be fully captured in our analysis.

There is an urgent need to develop and validate a disease-specific HRQoL measure tailored to children with respiratory illnesses in LMICs, with particular emphasis on the youngest age group. In the interim, existing HRQoL tools should be used cautiously, ensuring they are appropriately adapted to the socio-cultural context. Addressing these gaps will advance the understanding and assessment of HRQoL for children in LMICs, ultimately improving outcomes for those affected by respiratory illnesses, including TB.

## Supporting information

**S1 Appendix. Full electronic search strategy.**
(DOCX)

## Acknowledgments

We would like to thank the study staff and the support staff at Stellenbosch University, Desmond Tutu TB Centre for their dedication and assistance throughout the study.

## Author contributions

**Conceptualization:** Michaile Gizelle Anthony, Graeme Hoddinott, Marieke Margreet van der Zalm.

**Data curation:** Michaile Gizelle Anthony, Dzunisani Patience Baloyi.

**Formal analysis:** Michaile Gizelle Anthony, Graeme Hoddinott, Marieke Margreet van der Zalm.

**Funding acquisition:** Marieke Margreet van der Zalm.

**Methodology:** Michaile Gizelle Anthony, Graeme Hoddinott, Marieke Margreet van der Zalm.

**Project administration:** Michaile Gizelle Anthony.

**Resources:** Michaile Gizelle Anthony.

**Supervision:** Graeme Hoddinott, Marieke Margreet van der Zalm.

**Visualization:** Michaile Gizelle Anthony.

**Writing – original draft:** Michaile Gizelle Anthony.

**Writing – review & editing:** Graeme Hoddinott, Dzunisani Patience Baloyi, Anneke Catharina Hesseling, Marieke Margreet van der Zalm.

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
