## [Decision Letter · Decision Letter 0]

10 Oct 2024

PONE-D-24-06291Health-related quality of life measures for children: a scoping review of generic and respiratory illness-specific measuresPLOS ONE

Dear Dr. Anthony,

Thank you for submitting your manuscript to PLOS ONE. After careful consideration, we feel that it has merit but does not fully meet PLOS ONE’s publication criteria as it currently stands. Therefore, we invite you to submit a revised version of the manuscript that addresses the points raised during the review process.

Apart from all comments raised in the reviews, please make sure you especially take the following issues into account:  academic writing, please check the background section in relationship to the research questions, address the search strategy in relation to the comprehensiveness of the search results, and address the focus on TB in the abstract and background. 

We look forward to receiving your revised manuscript.

Kind regards,

Mathieu F. Janssen, Ph.D.

Academic Editor

PLOS ONE

Journal Requirements: When submitting your revision, we need you to address these additional requirements. 1. Please ensure that your manuscript meets PLOS ONE's style requirements, including those for file naming. The PLOS ONE style templates can be found at https://journals.plos.org/plosone/s/file?id=wjVg/PLOSOne_formatting_sample_main_body.pdf and https://journals.plos.org/plosone/s/file?id=ba62/PLOSOne_formatting_sample_title_authors_affiliations.pdf 2. We note that your Data Availability Statement is currently as follows: All relevant data are within the manuscript and its supporting information files. Please confirm at this time whether or not your submission contains all raw data required to replicate the results of your study. Authors must share the “minimal data set” for their submission. PLOS defines the minimal data set to consist of the data required to replicate all study findings reported in the article, as well as related metadata and methods (https://journals.plos.org/plosone/s/data-availability#loc-minimal-data-set-definition). For example, authors should submit the following data: - The values behind the means, standard deviations and other measures reported;- The values used to build graphs;- The points extracted from images for analysis. Authors do not need to submit their entire data set if only a portion of the data was used in the reported study. If your submission does not contain these data, please either upload them as Supporting Information files or deposit them to a stable, public repository and provide us with the relevant URLs, DOIs, or accession numbers. For a list of recommended repositories, please see https://journals.plos.org/plosone/s/recommended-repositories. If there are ethical or legal restrictions on sharing a de-identified data set, please explain them in detail (e.g., data contain potentially sensitive information, data are owned by a third-party organization, etc.) and who has imposed them (e.g., an ethics committee). Please also provide contact information for a data access committee, ethics committee, or other institutional body to which data requests may be sent. If data are owned by a third party, please indicate how others may request data access. 3. Please include your full ethics statement in the ‘Methods’ section of your manuscript file. In your statement, please include the full name of the IRB or ethics committee who approved or waived your study, as well as whether or not you obtained informed written or verbal consent. If consent was waived for your study, please include this information in your statement as well. 4. Please include captions for your Supporting Information files at the end of your manuscript, and update any in-text citations to match accordingly. Please see our Supporting Information guidelines for more information: http://journals.plos.org/plosone/s/supporting-information.

Reviewers' comments:

Reviewer's Responses to Questions

**Comments to the Author**

1. Is the manuscript technically sound, and do the data support the conclusions?

Reviewer #1: Partly

Reviewer #2: Partly

Reviewer #3: Partly

Reviewer #4: Yes

2. Has the statistical analysis been performed appropriately and rigorously? 

Reviewer #1: N/A

Reviewer #2: N/A

Reviewer #3: N/A

Reviewer #4: Yes

3. Have the authors made all data underlying the findings in their manuscript fully available?

Reviewer #1: Yes

Reviewer #2: Yes

Reviewer #3: Yes

Reviewer #4: Yes

4. Is the manuscript presented in an intelligible fashion and written in standard English?

Reviewer #1: No

Reviewer #2: Yes

Reviewer #3: Yes

Reviewer #4: Yes

5. Review Comments to the Author

Reviewer #1: Thank you for the opportunity to review the manuscript titled “Health-related quality of life measures for children: a scoping review of generic and respiratory illness-specific measures.” Development of a new respiratory HRQoL measure will strengthen the evidence base for this disease. The manuscript requires major revisions. I hope that the comments below will assist with this revision.

1. Abstract

a. Please specify the age of children in the abstract.

2. Background

a. Please justify why TB is specifically referenced in the abstract and Background. If the measure is designed for a range of respiratory illnesses then further background on this is justified. Currently TB is the focus. Although TB is important in the SA context this may limit the applicability of a future measure in other settings. I suggest that respiratory illness be given a working definition for this review.

b. “However, there is currently no respiratory disease-specific HRQoL measure for children.” Off the top of my head there are indeed other are other respiratory specific HRQoL measures available e.g. CFQ-R (CF), PedsQL Asthma, TACQoL Asthma, HAY . Please add this to your discussion and consider revision of the statement to imply general respiratory measures if that was implied.

c. Please justify why generic and disease-specific measures for 0-18-years were reviewed when the aim was to identify key components for children 0-5 years.

d. Kindly include what additional information your review will contribute considering the review by Quittner et al in 2008 (Quittner AL, Modi A, Cruz I. Systematic review of health-related quality of life measures for children with respiratory conditions. Paediatr Respir Rev. 2008 Sep;9(3):220-32. doi: 10.1016/j.prrv.2008.05.003. Epub 2008 Jul 26. PMID: 18694714.) and Kwon et al in 2023 (Kwon, Joseph, et al. "Systematic review of the psychometric performance of generic childhood multi-attribute utility instruments." Applied Health Economics and Health Policy 21.4 (2023): 559-584.)

3. Methods

a. Please justify the search terms further. Some important terms seem to be excluded e.g. outcomes, PROMs, respiratory etc

b. Was data limited to full-text articles?

c. Were the reference lists searched to identify additional publications?

d. Was the review registered in PROSPERO or another registry?

e. Please justify the age range of exclusion for ‘adolescents’

f. Please specify if software was used for data extraction and screening.

g. Please justify why “articles related to diseases other than respiratory illnesses were excluded.” I assume that the development of many generic HRQoL measures did not include respiratory illness.

h. Please define respiratory illness for the review. Was a study on children with a tracheostomy and measuring health on the PTHSI or with non-invasive ventilation with muscle weakness from a neuromuscular disease included or excluded?

i. Please specify what ‘Google search engine’ refers to.

j. What QoL websites were searched and/or how were they determined? What about conference proceedings.

k. Data extraction – who piloted the charting/data extraction table. Please specify domains – was this a domain which may have consisted of additional items e.g. Physical functioning or were domains and individual items extracted?

l. Please describe how age ranges were categorized for this manuscript.

4. Results

a. HRQOL Domains and items In general, the domains are reflective of the definition of HRQoL. Please justify why items were not reviewed by domain if your aim was to identify what would be important for a new measure.

b. I am not sure that your analysis of translated versions is correct – This may be included in studies that you have excluded. E.g. PedsQl generic measure is validated and translated into South African languages. Consider omitting data on translations.

c. The data extraction is currently limited and does not address consistency of use, definitions and criteria which are alluded to in the discussion.

5. Discussion

a. “There has also been 231 limited development or validation of available measures in LMIC, where the socio-economic 232 context and the burden of respiratory diseases is different.” Your study was not designed to discuss the validation of measures in LMIC. These may have been applied but were excluded in your review.

If the intention is to motivate for a new measure in a LMIC considering our burden of disease, then this needs to be discussed in more detail in the background. This is currently only related to TB. Consider adding discussion of the SDGs and other global health initiatives.

b. It is unclear how the considerations for age-specific developmental milestones is concluded from your results.

c. Please justify “most 243 HRQoL measures were originally developed in HICs and cannot be translated into other 244 settings without considering the different socioeconomic contexts”. In general guidelines suggest that there is cultural adaptation and translation when outcome measures are used in different contexts. This has been done successfully with many instruments.

d. Observable behaviours are included in your discussion. It is currently not obviously recorded in the results.

e. Further discussion points such as ‘a new measure should include extended social support networks and socioeconomic context and factors that are significant in LMICs’ are quite general and sweeping. This extraction of this data from the existing measures would strengthen the review.

6. General

a. Please consider revising for academic writing. I suggest that this includes less reference to ‘we’

Reviewer #2: I would like to thank the authors for their effort in addressing the important topic of HRQoL measures for children. This study, which attempts to chart available generic and respiratory disease-specific HRQoL measures, covers a significant area. However, I have several concerns and think there are several aspects require revision.

Major Concerns:

1. There appears to be a disconnect between the background and the research questions. While the research aim, as the title suggests, is to chart HRQoL measures for children broadly, the study's initial focus is on TB patients in low- and middle-income countries, which feels more specific. I feel either the authors need to revise the title and the research aims or they should provide more comprehensive context on the existing HRQoL measures—specifically, which are the most commonly used generic and respiratory illness-specific measures for children, and how have they been applied in previous research?

2. The author mentioned “there is currently no respiratory disease-specific HRQoL measure for children”, I do not think this is true, as this study was about to identifying these measures.

3. It is also unclear to me that, how non-TB disease-specific instruments will contribute to assessing HRQoL in TB patients. The authors may need to either expand in the background to explain why they included respiratory illness instruments in general or narrow their focus to TB-specific or generic instruments that are used in TB patients.

4. I have significant concerns about the search strategy employed, which casts doubt on the comprehensiveness of the results. the authors should provide the search terms in the appendix for transparency. Given the broad scope of the search, it is surprising that only 1,823 papers were identified. This relatively small yield suggests the search may not have been as exhaustive as intended. I recommend that the search strategy be re-evaluated to ensure it is capturing all relevant literature.

5. The exclusion criteria, particularly for generic measures, are unclear. It is not clear how the authors can identify generic measures for use in children with respiratory illness, without looking at those paper “assessing the quality of life using the measure”. A clearer outline of these criteria is necessary to justify the results.

6. Upon reviewing the keywords, it seems no terms specifically related to respiratory illnesses were included. This omission likely reduces the search’s effectiveness in capturing relevant studies. Additionally, the use of terms like "develop" and "validation" may have inadvertently excluded valuable studies that assess instruments but do not focus on their development or validation.

7. the authors limited their search to studies published between January 2000 and November 2023. Many widely used generic HRQoL measures were developed before 2000, so this time restriction may exclude important foundational studies. I therefore have major concern on this timeframe.

8. The manuscript does not include quality assessment of the reviewed studies or HRQoL instruments. Have lall the included instruments been validated? A formal quality assessment framework is essential in systematic reviews.

9. It would be helpful to include a frequency chart showing how often each domain (e.g., physical, emotional, social functioning) is covered by the identified generic and disease-specific instruments. This would allow readers to better understand which areas of HRQoL are well-represented and mostly related to the diseases.

10. Providing detailed information about the specific items included in the various instruments would further improve the manuscript’s utility. This would give readers a better understanding of which aspects of HRQoL are being measured by each instrument, and could help guide future research or clinical applications.

Reviewer #3: This study addresses a significant gap in the literature regarding HRQoL assessment for children affected by respiratory illnesses in a low and middle income countries. While the authors have raised an important issue, I believe the execution could be improved. Therefore, I have suggestion in order to improve the structure and the readability of the paper.

1. Clarity of the background: In general, I think the authors need to organizing the background more clearly so that the reader can understand the depth and the impact of your study. As it stands, the rationale for the study is challenging to follow, which may hinder readers’ understanding of its significance.

2. I question the authors statement that

“our understanding of the impact of respiratory illnesses, including TB, on the HRQoL of children, both during the disease episode and beyond treatment completion, is extremely limited.”

There is a considerable body of research on TB, asthma, and other respiratory illnesses. The authors should provide a stronger argument for why a scoping study is necessary to develop a new tool in this field.

3. What do you mean by ‘there is no respiratory disease specific hrqol measure for children’? In your study itself, you have 21 studies about this topic.

Line 127

4. What do you mean by QoL websites? Is there any?

5. What is the authors consideration by using Google scholar as the primary tool for literature searches?

6. Do you think involving grey literature may affect the quality of your study? Please put this in the discussion section.

7. Line 160

What do you mean by discrete measures?

8. Table 2:

a. Please put the table heading on each page

b. Remove the phrase ‘years old’

c. Decrease the font size for better fit

d. Remove the terms ‘domain’, and ‘items’

e. Remove the phrase ‘Measurement’, retain the scale

f. Add a column to indicate whether the measure has been validated in the LMIC (yes/no)

DISCUSSION

9. IN the first paragraph, authors should summarize the main findings that address the research question. In your study, I suppose that the study aim is the key component of HRQoL instrument for children.

Reviewer #4: Thank you for inviting me to review this study. I think this is a valuable work that could contribute to this field. I have several major comments for the authors to consider. Also, consider proofreading this manuscript before publication.

1. While I support researchers to develop new instrument, it is important to acknowledge the limitations of existing instruments. In other words, if no studies have demonstrated the lack of validity and sensitivity of existing instruments, why would the authors consider developing a new instrument in first place? On the one hand, a newly developed instrument may have a better sensitivity, on the other hand, this may hamper comparison across studies if everyone is using a different instrument.

2. While the manuscript provides a detailed list of HRQoL measures and their dimensions, there is some inconsistency in defining and categorizing domains across different instruments. Consider using a standardized framework to group these dimensions for easier comparison.

3. The manuscript highlights the need for a new HRQoL tool for young children in LMICs but does not provide specific guidance on how this new tool should differ from existing ones. Including a discussion on specific attributes (e.g., developmental milestones, socio-economic context, or caregiver input) that should be emphasized would strengthen the practical utility of the review. Also, I am not sure about how a HRQoL tool in LMICs could differ with the HRQoL tool used in developed countries. Can you clarify?

4. The exclusion of studies published before 2000 and those not in English is reasonable given resource constraints. However, briefly acknowledging potential biases or limitations this might introduce would add transparency to the review process. This could be a big limitation considering the aim is to develop a HRQoL tool for LMICs. I could imagine many studies from LMICs are published in non-English journals?

5. In the key words used for this scoping review, I do not see the word “LMICs”. This seems inconsistent between the scoping review and the motivation of conducting this study.

6. PLOS authors have the option to publish the peer review history of their article (what does this mean? ). If published, this will include your full peer review and any attached files.

**Do you want your identity to be public for this peer review?** For information about this choice, including consent withdrawal, please see our Privacy Policy .

Reviewer #1: No

Reviewer #2: No

Reviewer #3: No

Reviewer #4: **Yes: ** Zhihao Yang

---

## [Author Response · Author response to Decision Letter 0]

23 Nov 2024

I have attached a point by point response to the reviewers comments

---

## [Decision Letter · Decision Letter 1]

2 Feb 2025

PONE-D-24-06291R1Health-related quality of life measures for children: a scoping review of generic and respiratory illness-specific measuresPLOS ONE

Dear Dr. Anthony,

Thank you for submitting your manuscript to PLOS ONE. After careful consideration, we feel that it has merit but does not fully meet PLOS ONE’s publication criteria as it currently stands. Therefore, we invite you to submit a revised version of the manuscript that addresses the points raised during the review process.

We look forward to receiving your revised manuscript.

Kind regards,

Mathieu F. Janssen, Ph.D.

Academic Editor

PLOS ONE

Journal Requirements:

**Additional Editor Comments:**

As reviewer #1 points out, the manuscript has improved substantially, but there are still quite a few problematic issues to address. Please take all comments raised by the reviewer into account and revise accordingly. 

Reviewers' comments:

Reviewer's Responses to Questions

**Comments to the Author**

1. If the authors have adequately addressed your comments raised in a previous round of review and you feel that this manuscript is now acceptable for publication, you may indicate that here to bypass the “Comments to the Author” section, enter your conflict of interest statement in the “Confidential to Editor” section, and submit your "Accept" recommendation.

Reviewer #1: (No Response)

Reviewer #3: (No Response)

2. Is the manuscript technically sound, and do the data support the conclusions?

Reviewer #1: Partly

Reviewer #3: (No Response)

3. Has the statistical analysis been performed appropriately and rigorously? 

Reviewer #1: N/A

Reviewer #3: (No Response)

4. Have the authors made all data underlying the findings in their manuscript fully available?

Reviewer #1: Yes

Reviewer #3: (No Response)

5. Is the manuscript presented in an intelligible fashion and written in standard English?

Reviewer #1: Yes

Reviewer #3: (No Response)

6. Review Comments to the Author

Reviewer #1: Thank you for the opportunity to review this revised manuscript. I acknowledge the immense work that the authors have put into these revisions. Considering my previous comments and the comments of the other four reviewers I believe that this manuscript still requires substantial revisions.

1. I agree with reviewer 2 that the title and aim of the systematic review could be changed to better clarify your aim. It appears that the aim of your review is in fact to identify candidate items to include in a new measure of (HRQoL) for children living with respiratory illness in a LMIC setting. I suggest that your review is revised to clarify this and the results and discussion should better reflect this.

This review has not critically appraised the measures or assessed their suitability for use in a LMIC nor within general respiratory illness.

2. Despite the authors stating that the reference to TB had been removed from the abstract, it in fact has not.

3. The updated text justifying the need to specifically reference TB has lead to further ambiguities.

3.1 The authors second sentence states that Viral infections account for 90% of all paediatric LRTIs which, to me, would imply that this would be the most important consideration for developing a respiratory HRQoL measure.

3.2 Line 75-77 state that only half of the 1.3 million children received care with the diagnostic gap largest in those < 5 years – this indicates a need for improved diagnostic and treatment packages and not necessarily a HRQoL measure.

3.3 The emphasis on TB is still incredibly large considering that this measure is intended to be used as a general measure for a variety of respiratory illnesses.

4. Line 84-86 requires a reference.

5. Line 86-89 requires a reference.

6. Reference 20 can be used to describe the challenges of decreased energy, sleep disturbances and social stigma but NOT that no existing HRQoL measure addresses these issues. A new reference should be added to defend the second half of this sentence.

7. Please add a reference for line 104-105 “they are often overly specific and not suitable for general use across respiratory illness”

8. Please add a reference to substantiate the argument in line 105-107.

9. Please expand on what is meant by ‘critical gaps’

10. If the aim of your research is to identify items that may be relevant to include in the development of a new measure then I would suggest that this be made clearer.

11. Inclusion criteria states that articles that evaluated the psychometric properties of HRQoL measures was included – this is not evident from your results or reference list.

12. Previous comment 7 : Methods – was data limited to full-text articles has not been addressed. Authors response “This rigorous screening process ensured the inclusion of studies relevant to QoL in respiratory illnesses among children. Both published peer reviewed articles and grey literature were included to provide a more comprehensive understanding of existing evidence.” Does not refer to full-text articles nor is this included in the inclusion/exclusion criteria.

13. Your response to previous comment 13 is appreciated. It is however not addressed in the manuscript. Please ensure that these details are captured in the text or in your table of inclusion and exclusion criteria.

14. Findings line 295 please specify the four generic measures identified for children aged 0-5 years.

15. Findings – kindly note that the TANDI has been renamed the EQ-TIPS and should be referenced as such.

16. Findings age range – the PedsQL has developed specific versions for children aged between 0-5 years. The TAPQoL, ITQol, C-QoL, HSCS-PS (HuPs). Please refer to Kwon – I believe that there are close to 20 versions available for children <5 years.

17. Line 338 “One measure did not report 338 specific domains” please specify what measure this is.

18. Line 368. The development of the EQ-5D-Y took place in multiple countries, including SA, it was not just validated in a LMIC.

19. Discussion line 420 – “There is a paucity of HRQoL measures—both generic and disease-specific—specifically designed for children aged 0-5-years, a group disproportionately affected by respiratory illnesses”

This statement is quite controversial and I do not think that it is a finding of your study. As you did not include the performance of generic measures in respiratory disease or appraise the psychometric performance of these measures in respiratory disease nor LMIC nor 0-5 years of age.

20. Existing tools exhibit inconsistent definitions, criteria, and inclusion of domains and items, limiting comparability across studies. Your results section does not address the definitions applied to these measures. Your study did not appraise the domains, items and comparability of instrument findings.

21. I agree that most HRQoL measures were developed in HICs. This review cannot substantiate that these measures are not readily adaptable to LMICs without accounting for distinct cultural, linguistic and socioeconomic factors. Your search criteria nor your results allow this discussion nor this conclusion.

22. Lines 433-439 and 440-442 are important to note. It is not clear that this was the aim nor the focus of this literature review.

Reviewer #3: The authors have provided a comprehensive answer to my question. I appreciate their effort in revising the manuscript. I have no further comments.

7. PLOS authors have the option to publish the peer review history of their article (what does this mean? ). If published, this will include your full peer review and any attached files.

**Do you want your identity to be public for this peer review?** For information about this choice, including consent withdrawal, please see our Privacy Policy .

Reviewer #1: No

Reviewer #3: No

---

## [Author Response · Author response to Decision Letter 1]

5 Mar 2025

I have attached a revised response based on the recommendation to submit the manuscript without track changes to reviewers and the revised manuscript (track changes accepted)

---

## [Editor Report · Decision Letter 2]

24 Mar 2025

Identifying candidate items for a HRQOL measure in young children with respiratory Illness: A scoping review of generic and disease-specific measures

PONE-D-24-06291R2

Dear Dr. Anthony,

We’re pleased to inform you that your manuscript has been judged scientifically suitable for publication and will be formally accepted for publication once it meets all outstanding technical requirements.

Kind regards,

Mathieu F. Janssen, Ph.D.

Academic Editor

PLOS ONE
---

## [Editor Report · Acceptance letter]

PONE-D-24-06291R2

PLOS ONE

Dear Dr. Anthony,

I'm pleased to inform you that your manuscript has been deemed suitable for publication in PLOS ONE. Congratulations! Your manuscript is now being handed over to our production team.

Kind regards,

on behalf of

Dr. Mathieu F. Janssen

Academic Editor

PLOS ONE